

# The importance of dung beetles and arthropod communities on degradation of cattle dung pats in eastern South Dakota

Jacob R. Pecenka[1,2] and Jonathan G. Lundgren[2]

[1] Natural Resource Management Department, South Dakota State University, Brookings, SD, United States of America
[2] Ecdysis Foundation, Estelline, SD, United States of America

## ABSTRACT

**Background**. Dung accumulation in rangelands can suppress plant growth, foul pastures, and increase pest pressure. Here, we describe the arthropod community of dung in eastern South Dakota, and quantify their contributions to dung degradation using an exclusion cage design.
**Methods**. Various arthropod community and degradation characteristics were measured in caged and uncaged dung pats over time in early and late summer.
**Results**. A total of 86,969 specimens were collected across 109 morphospecies (13 orders) of arthropods, and cages effectively reduced arthropod abundance, species richness, and diversity. Uncaged dung pats degraded significantly faster than the caged pats, with the largest difference occurring within 2 d of pat deposition. Dung organic matter was degraded more slowly (by 33–38 d) in the caged pats than where insects had free access to the pats. Although dung beetles only represented 1.5–3% of total arthropod abundance, they were significantly correlated to more abundant and complex total arthropod communities.
**Discussion**. A diverse community contributes to dung degradation in rangelands, and their early colonization is key to maximizing this ecosystem service.

## INTRODUCTION

When cattle excrete dung onto the soil surface, the failure of the pats to break down can challenge the productivity of grazing on rangelands (*Fincher, 1981*). When cattle consume forage, any nutrients not digested are returned to the system in the form of dung and urine (*Haynes & Williams, 1993*; *Wu & Sun, 2010*). The undigested plant material that comprises dung is deposited on the soil surface, smothering plant growth in that area (*Holter, 2016*; *MacLusky, 1960*). Pasture fouling through continuous dung deposition that fails to degrade quickly can represent a substantial problem to ranchers if left unmanaged. When a dung pat is deposited on a pasture, all of the available forage underneath and up to a 5 m radius around the pat is unused by grazing cattle until the pat is incorporated into the soil

Corresponding author
Jonathan G. Lundgren,
jgl.entomology@gmail.com

(*Weeda, 1967*). Dung loses 22% (but up to 80%) of its nitrogen (N) to volatilization within 60 d of deposition (*Nichols et al., 2008*; *Weeda, 1967*). Other important nutrients such as phosphorus (P) and potassium (K) are present in dung pats in much smaller quantities and can be lost to leaching and runoff when left on the soil surface (*Gillard, 1967*; *Nichols et al., 2008*; *Petersen, Lucas & Woodhouse, 1956*). Volatilization and reductions of these elements decreases the nutrient availability to the plant community, resulting in lower quantity and quality of forage for future cattle grazing (*Aarons et al., 2009*; *Bang et al., 2005*).

A variety of factors affect how quickly dung is incorporated into the soil. Reports of dung degradation rates vary from 50–65 d over the season (*Holter, 1979*), 57–78 d in late spring to 88–111 d in late summer (*Lee & Wall, 2006*), and up to 3 y in cattle grazing systems with high insecticide use (*Anderson, Merritt & Loomis, 1984*; *Strong, 1992*). This variability is due to factors that include weather (*Holter, 1979*), seasonality (*Lee & Wall, 2006*), insecticide use (*Suarez et al., 2003*), and the nutritional quality of the dung itself (*Cook, Dadour & Ali, 1996*). Degradation of dung pats is facilitated by arthropods that accelerate the incorporation of the dung pat organic matter into the soil, and improve soil's aeration and water holding capacity (*Macqueen & Beirne, 1975*). Dung often supports dozens or even hundreds of arthropod species (*Blume, 1985*; *Merritt & Anderson, 1977*; *Valiela, 1969*).

Arthropods that colonize dung pats can be categorized into different functional guilds that each contribute to the eventual incorporation of the dung into the soil. One of the first studies that considered dung community function was *Mohr (1943)*, which prompted other studies that documented arthropod succession in a dung pat and their varying niches within the micro-habitat (*Cervenka & Moon, 1991*; *Koskela & Hanski, 1977*; *Sanders & Dobson, 1966*). These studies are accompanied by more recent explorations of the importance of dung beetles (Scarabaeidae) and the multiple ecosystem services that they provide (*Beynon et al., 2012*; *Manning et al., 2016*; *Nichols et al., 2008*). One of these ecosystem services is dung beetles' ability to increase the productivity of rangeland ecosystems (*Bang et al., 2005*; *Penttila et al., 2013*). Increased rangeland productivity is achieved by bioturbation and burial of dung that results in capturing ephemeral nutrients for surrounding forage plants up to 12.7 cm away from the dung pat (*Bornemissza, 1970*; *Macqueen & Beirne, 1975*; *Yamada et al., 2007*). A second important ecosystem service is the suppression of dung inhabiting pests to grazing cattle (*Fincher, 1981*). By removing nutritional resources and habitat, dung beetles reduce pest maggot abundance (*Doube, 1990*; *Nichols et al., 2008*). Suppressing these pests is accelerated when natural enemies such as predatory staphylinid or hister beetles and parasitoid wasps colonize the dung pat (*Cervenka & Moon, 1991*). These ecosystem services provided by dung beetles (as well as other members of the dung arthropod community) have an economic value to the ranching operation (*Beynon, Wainwright & Christie, 2015*), but most of the numbers used to generate these values are at least 37 years old (*Beynon et al., 2012*; *Fincher, 1981*; *Losey & Vaughan, 2006*).

The goal of our study was to document the dung insect community in eastern South Dakota, and determine dung degradation rates over time in the presence and absence of this community. Cages like those employed here help to isolate the contribution of
the majority of the arthropod community to dung pat degradation (*Lee & Wall, 2006*; *Tixier, Lumaret & Sullivan, 2015*). Here, we pair cages with a comprehensive description of invertebrate communities within the dung both early and late in the summer to understand how elements of this community affect degradation over the season. There are 9.12 million ha of rangeland in South Dakota (USDA-NASS), and this region represents an important transition zone between the mid and tall-grass prairie biomes (*NASS, 2016*). Dung-inhabiting Coleoptera from South Dakota were described nearly 50 years ago (*Kessler & Balsbaugh, 1972*; *McDaniel, Boddicker & Balsbaugh, 1971*), but these studies did not correlate these insects to dung pat degradation, and land use patterns have changed dramatically toward annual cropland over this period of time (*Johnston, 2014*; *Wright & Wimberly, 2012*). Identifying the impact of the dung arthropod community on degradation will provide ranchers a greater understanding of the benefits of conserving this poorly understood community. We hypothesize that caging pats will reduce insect colonization of the dung and impede pat degradation rates.

## MATERIALS AND METHODS

### Study site

This study was conducted on a ranch in eastern South Dakota, US, at 44.758, −96.538 in the summer of 2016. The study site was at an altitude of 559 m in an area with an average annual rainfall of 684 mm and an average summer temperature of 19.8 °C. The 130 ha pasture was composed of mixed grasses; consisting mostly of *Schizachyrium scoparium* (little bluestem), *Andropogon gerardii* (big bluestem) and *Spartina pectinata* (prairie cordgrass) with predominantly silty clay and silty clay loam soil types (*USDA-NRCS, 2016*). The grazing season prior to and during the experiment had a 130-steer herd made up of Angus, Belted Galloway, and Irish Black breeds, that was moved among small 0.41–1.21 ha paddocks approximately every 24 h. Cattle were excluded from the experimental site during the observation periods. No insecticide or nematicide treatments had been used on cattle on this ranching operation in more than 10 y.

### Dung degradation measurements

Dung (<2 h old; 90 kg collected twice) was collected from the pasture on 04/06/2016 and 05/06/2016 before 10:00. Fresh dung pats were homogenized and stored in bags at −25 °C for 72 h to ensure all arthropods had been killed. Dung was removed from the freezer, completely thawed, and homogenized prior to use in the experiment. Aliquots of the dung (1,000 ± 10 g) were weighed, individually bagged, and stored for 24 h before placing them in the field. Each bag of weighed dung became a "sentinel pat" to represent a dung pat deposited by grazing cattle. Observation sites ($n = 84$) were placed in the pasture so sites were at least 5 m apart. At each site, a sentinel pat was placed on top of mesh with 2.5 cm square holes to allow for ease of pat removal. Each site was randomly assigned to one of three treatments. In the first treatment (inclusion; $n = 36$), dung pats were left completely exposed with no covering. In the second treatment (exclusion; $n = 36$), the pats were surrounded by a PVC cylinder (25 cm diam., 25 cm tall), buried at least 12 cm into the ground to reduce ground colonization of the pat. The tops of these cylinders were covered
in fine mesh screen (<1 mm opening) and secured with a plastic tie. The final control treatment (sham cage; $n = 12$) used the same cylinder design as the exclusion treatment, but with three $10 \times 10$ cm holes cut on the sides to allow arthropods to travel into the cylinder. A wire top was used to cover these sham cylinders that had 3 cm openings. This third treatment was added to test whether the exclusion cage had direct effects on dung degradation rates.

To determine degradation rates, pats from the three treatments were weighed over time. The entire experiment was repeated twice over the season, once beginning on 10-June and once on 28-July. Randomly selected pats (Inclusion [six pats], Exclusion cage [six], and Sham cage [two] on each time point) were removed 2, 4, 7, 14, 28, and 42 d after the sentinel pats were placed. At the time of removal, the pat was collected in a plastic bag, sealed and taken to the laboratory. Each pat was weighed while still fresh and after drying to constant weight (over 7–10 d). A 10-gram sample of this dried pat was ground to a fine powder and baked at 500 °C for 1 h; the remaining sample was then re-weighed to determine ash/mineral content of the sample. From this value the ash-free organic matter content (AFOM) of the pat was calculated.

## Arthropod collection and dung pat analysis

The pat was weighed and placed in a Berlese funnel system for 7 d to extract arthropods living in the dung pat. The top of the extraction funnel sealed with the top board of this extraction system, eliminating the ability of winged arthropods to escape. The arthropods were identified under a microscope and then weighed to calculate arthropod biomass. To help characterize this diverse arthropod community, each specimen was identified to the lowest taxonomic level possible. Specimens were identified to at least the family level using *Triplehorn & Johnson (2005)*; and scarabid beetles were identified to species by *Ratcliffe & Paulsen (2008)*. Within these families, each specimen was assigned to a morphospecies and functional guild depending on their feeding ecology. The non-pest coprophagous community was divided into macro-coprophages (>1 mm long; Scarabaeidae, Hydrophilidae), and micro-coprophages (<1 mm long; Acarina, Collembola, Ptiliidae).

## Data analysis

All statistics were conducted using Systat 13 (SYSTAT Software, Inc; Point Richmond, CA). Two-way ANOVAs were used to investigate how dung pat age and cages affected dung pat and arthropod characteristics including pat wet weight, dry weight, moisture content, organic matter content, and arthropod biomass, abundance, species richness (number of morphospecies found), species diversity (Shannon H), and abundance of family Scarabaeidae. To avoid possible pseudoreplication and sampling bias, separate analyses were conducted on data collected early and late in the season. ANOVAs were used to compare the abundances of coprophages, predators, parasitoids, herbivores, and maggots collected per pat early and late in the season. Linear regressions were generated to compare the number of dung beetles to dung pat organic matter, total arthropod biomass, arthropod abundance, species richness, species diversity and micro-coprophage abundance

(pooled across pat ages and early/late season observations). Many cross comparisons of different community characteristics and response variables can lead to false positives, or type I errors, in our analyses. Prior to running any statistical tests, we always investigate the patterns in the data, looking for biologically meaningful trends. This helps to reduce the likelihood of type I errors and increase the relevance of our results.

## RESULTS

### Dung arthropod community

A total of 109 morphospecies (86,969 arthropod specimens) were collected from dung pats, representing 13 orders (Acarina, Araneae, Coleoptera, Collembola, Diptera, Hemiptera, Hymenoptera, Isopoda, Julida, Lepidoptera, Lithobiomorpha, Pseudoscorpiones, and Thysanoptera). There were $517.68 \pm 30.98$ (mean $\pm$ SEM) larval and adult specimens collected ($228.02 \pm 46.05$ mg of arthropods), represented by $13.32 \pm 0.49$ morphospecies, per dung pat. Larval communities included only three orders; Diptera (six morphospecies), Coleoptera (17 morphospecies) and Lepidoptera (one morphospecies). Orders with the most abundant specimens were Acarina ($n = 35,534$), Coleoptera (adult $n = 22,689$; larvae $n = 6,057$), Collembola ($n = 9,114$), Diptera (adult $n = 609$; larvae $n = 8,870$), Lepidoptera (adult $n = 8$; larvae $n = 2,034$), and Hymenoptera ($n = 1,141$). Four families of Coleoptera were well represented (they comprised 26% of all specimens collected): Staphylinidae ($n = 8,140$), Ptiliidae ($n = 9,247$), Hydrophilidae ($n = 3,576$), Scarabaeidae ($n = 1,624$) were represented by 14, one, 12, and 13 morphospecies from these families, respectively. Trophically, these specimens were categorized as coprophagous (37 morphospecies; 60,564 specimens), predators (38 morphospecies; 15,047 specimens), herbivores (18 morphospecies; 2,037 specimens;) or parasitoids (10 morphospecies; 539 specimens). The remaining specimens are regarded as coprophagous maggots (6 morphospecies; 8,870 specimens), consisting of Diptera larvae.

Arthropods were collected early and late in the summer. Arthropod abundances in the early season were $485.63 \pm 42.18$ (40,765 total) specimens from $12.91 \pm 0.71$ morphospecies per pat; $549.73 \pm 45.36$ (46,269 total) arthropods representing $13.73 \pm 0.67$ morphospecies per pat were collected later in the summer. Arthropod biomass per pat was $357.93 \pm 40.98$ mg in the early season and $152.72 \pm 72.40$ mg in the late season. Functional group populations changed between the two sampling periods. Coprophage abundance significantly ($F = 5.08$; $df = 1,166$; $P = 0.026$) increased 26% (25,837 to 34,639), predator abundance significantly ($F = 9.11$; $df = 1,166$; $P = 0.003$) increased 40% (5,661 to 9,386), parasitoid abundance significantly ($F = 24.36$; $df = 1,166$; $P < 0.001$) decreased by 78% (442 to 97), herbivore abundance significantly ($F = 7.89$; $df = 1,166$; $P = 0.012$) decreased by 53% (1,389 to 648), and maggots significantly ($F = 12.25$; $df = 1,166$; $P = 0.001$) decreased by 79% (7,371 to 1,499) from early to late summer. Dung beetles represented 3% and 1.5% of total arthropod abundance in early and late season, respectively.

### Sham cage effect

The sham cages had similar arthropod communities and dung characteristics with the no cage treatment in 16 of the 18 ANOVAs of different dung arthropod community groups

and dung degradation metrics. Only early season arthropod abundance and late season dung pat wet weight were significantly different between cage treatments, but these trends were not consistent in both early and late seasons. The general lack of differences between the sham cage and no cage treatments indicates that the cage had little direct effect on arthropod communities and dung characteristics, and justifies our focus on cage/no cage comparisons for the remainder of this section.

### Treatment and time effect on dung community

Dung pats in the pasture had different arthropod communities when pats were caged and as the dung pat aged. Cages and time had a significant effect on arthropod biomass in the early and late seasons (Fig. 1). Arthropod biomass inside the cages was 10% of the biomass found in the uncaged pats early in the season and caged pats had 13% of the biomass of the uncaged later in the summer. The biomass and abundances (Fig. 2) of arthropods were significantly greater on younger pats (2, 4 and 7 d old) versus older pats (14, 28, 42 d old) both early and late in the season. Specifically, the biomasses declined by 72 and 83% between the 7th and 14th days in early and late season, respectively. After 14 d, arthropod biomass did not significantly change through 42 d. Caged dung pats only averaged 52% and 57% of the arthropod specimens that were found in the inclusion dung pats in early and late seasons, respectively.

Cages did not completely exclude the insect community, but it did reduce the arthropod species richness and diversity. The richness of arthropod species found in the dung pats was significantly affected by cages and time in the early season (exclusion: $F = 102.86$; $df = 1,60$; $P < 0.001$; time: $F = 5.08$; $df = 5,60$; $P < 0.001$; interaction: $F = 6.71$; $df = 5,60$; $P < 0.001$) and in the late season (exclusion: $F = 66.49$; $df = 1,60$; $P < 0.001$; time: $F = 26.44$; $df = 5,60$; $P < 0.001$; interaction: $F = 6.70$; $df = 5,60$; $P < 0.001$). The mean number of species in the caged dung pats were 46% and 62% of the number in the uncaged pats for the early and late season, respectively. Cages and time had a significant effect on arthropod diversity (Shannon H) in the early season (cage: $F = 34.24$; $df = 1,60$; $P < 0.001$; time: $F = 10.71$; $df = 5,60$; $P < 0.001$; interaction: $F = 5.03$; $df = 5,60$; $P = 0.001$) but only time had a significant effect in the late (cage: $F = 0.40$; $df = 1,60$; $P = 0.528$; time: $F = 50.19$; $df = 5,60$; $P < 0.001$; interaction: $F = 5.53$; $df = 5,60$; $P < 0.001$) season. There were significantly more maggots ($F = 47.37$; $df = 2,66$; $P < 0.001$) in the caged than the uncaged pats. Dung beetle abundance was significantly reduced by the arthropod exclusion. In the early season, cages and time had a significant effect on dung beetle abundance (cage: $F = 111.55$; $df = 1,60$; $P < 0.001$; time: $F = 17.30$; $df = 5,60$; $P < 0.001$; interaction: $F = 17.95$; $df = 5,60$; $P < 0.001$). Likewise, in the late season cages and time had a significant effect on dung beetle abundance (cage: $F = 105.01$; $df = 1,60$; $P < 0.001$; time: $F = 23.85$; $df = 5,60$; $P < 0.001$; interaction: $F = 23.85$; $df = 5,60$; $P < 0.001$).

### The effects of arthropod reduction and time on dung degradation

Cages and time had significant effects on dung pat wet weight in both the early and late season (Table 1). Dung from which many insects were excluded had an average of 26.10

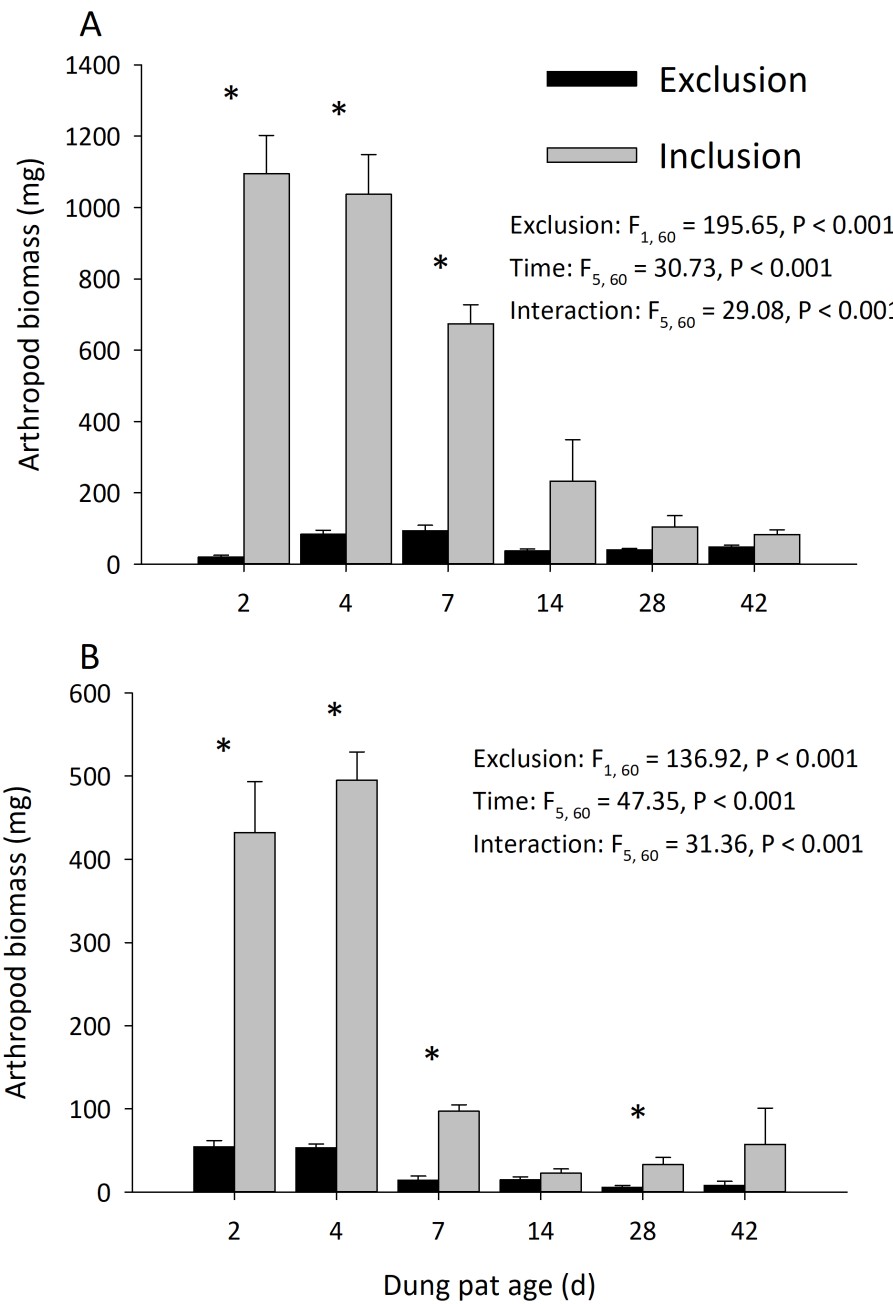

**Figure 1** **Arthropod dry weight biomass (mean ± SEM) per cattle dung pat ($n = 6$) over the age of the pat.** "Exclusion" refers to caged pats; "Inclusion" refers to uncaged pats. Arthropods were excluded from half of the pats ($n = 6$ pats per treatment per age) using cages. Pats were examined beginning in June (A) and in late July (B). Asterisks above the bars indicate significantly different arthropod biomasses in the caged and uncaged pats for that specific sample age ($\alpha = 0.05$).

and 21.93% lower wet weights and dry weights (in the early and late seasons) (Table 1) than when arthropods were allowed access to the pats. Pats experienced a $28.3 \pm 1.94\%$ wet weight loss during the first 2 d and $79.70 \pm 1.28\%$ weight loss by day 42 (Table 1). Moisture of dung pats was significantly correlated with arthropod abundance in early ($F_{1,70} = 23.59$,

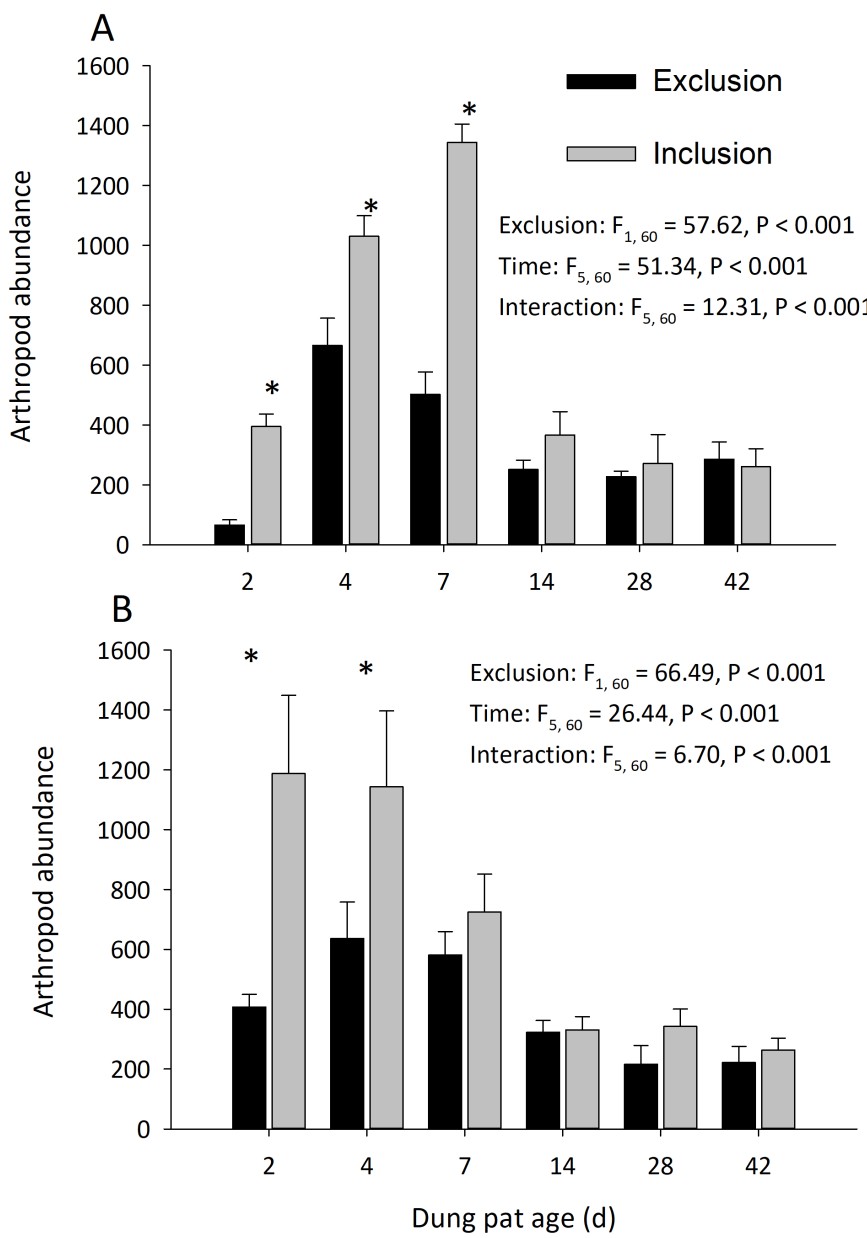

**Figure 2** **Arthropod abundance (mean ± SEM) per cattle dung pat ($n = 6$) over the age of the pat.**
"Exclusion" refers to caged pats; "Inclusion" refers to uncaged pats. Arthropods were excluded from half
of the pats ($n = 6$ pats per treatment per age) using cages. Pats were examined beginning in June (A) and
in late July (B). Asterisks above the bars indicate significantly different arthropod abundances in the caged
and uncaged pats for that specific sample age ($\alpha = 0.05$).

$P < 0.001$) and late ($F_{1,70} = 9.99$, $P = 0.002$) seasons. After drying, the uncaged dung pats
contained 25.45% and 25.10% less weight than the caged dung pats.

The ash-free organic matter (AFOM) percentage of the dried dung pats' remaining
weight was significantly affected by exclusion cages and time in the early and late seasons
(Table 1). Throughout the season the uncaged dung pats had significantly less AFOM

**Table 1  The effects of age in days since deposition (0–42 days) on of the weight and ash-free oganic matter of dung pats.** These characteristics were studied over time when insects were allowed access to (inclusion) or were excluded from the pats using cages. Communities were sampled early in the season and late in the season, and are presented distinctly. Data presented represents the mean ± SEM. Capital letters represent differences over time and lower case letters represent differences between treatments ($\alpha = 0.05$).

| | Days | Wet weight of pat (g) | | Dry weight of pat (g) | |
|---|---|---|---|---|---|
| | | Exclusion cage | Uncaged | Exclusion cage | Uncaged |
| Early season | 2 | 772.78 ± 24.34 Aa | 585.11 ± 30.12 Ab | 193.05 ± 4.18 Aa | 147.38 ± 12.10 Ab |
| | 4 | 697.25 ± 15.58 Ba | 521.28 ± 29.21 ABb | 144.91 ± 5.97 Ba | 103.14 ± 7.14 BCb |
| | 7 | 648.55 ± 24.70 Ba | 497.95 ± 18.77 Bb | 135.5 ± 2.76 BCa | 118.30 ± 6.68 BCb |
| | 14 | 359.96 ± 17.28 Ca | 261.94 ± 28.36 Cb | 156.93 ± 3.50 Ba | 103.87 ± 10.77 BCb |
| | 28 | 290.15 ± 18.59 Da | 215.04 ± 9.89 Cb | 135.75 ± 3.24 BCa | 99.55 ± 7.96 BDa |
| | 42 | 193.22 ± 15.92 Ea | 133.88 ± 5.40 Db | 115.41 ± 2.04 Da | 84.19 ± 3.41 Da |
| | | exclusion: $F_{1,60} = 104.21, P < 0.001$ time: $F_{5,60} = 207.96, P < 0.001$ interaction: $F_{5,60} = 3.31, P = 0.010$ | | exclusion: $F_{1,60} = 94.07, P < 0.001$ time: $F_{5,60} = 194.32, P < 0.001$ interaction: $F_{5,60} = 8.86, P < 0.001$ | |
| Late season | 2 | 829.21 ± 14.04 Aa | 708.13 ± 13.42 Ab | 301.64 ± 5.62 Aa | 237.64 ± 2.86 Ab |
| | 4 | 702.10 ± 9.57 Ba | 535.84 ± 10.21 Bb | 217.66 ± 6.64 Ba | 186.75 ± 1.55 Bb |
| | 7 | 741.68 ± 10.95 Ba | 630.58 ± 7.10 Cb | 220.30 ± 2.23 Ba | 187.11 ± 2.11 Bb |
| | 14 | 712.34 ± 13.14 Ba | 605.70 ± 9.85 Cb | 227.27 ± 6.99 Ba | 169.89 ± 3.89 Cb |
| | 28 | 442.28 ± 10.97 Ca | 274.27 ± 10.31 Db | 191.33 ± 2.80 Ca | 115.69 ± 4.09 Db |
| | 42 | 293.96 ± 9.47 Da | 219.38 ± 8.55 Eb | 181.30 ± 3.79 Ca | 117.21 ± 4.52 Db |
| | | exclusion: $F_{1,60} = 398.59, P < 0.001$ time: $F_{5,60} = 703.00, P < 0.001$ interaction: $F_{5,60} = 5.68, P < 0.001$ | | exclusion: $F_{1,60} = 482.79, P < 0.001$ time: $F_{5,60} = 208.54, P < 0.001$ interaction: $F_{5,60} = 9.02, P < 0.001$ | |

| | Days | Ash free organic matter weight of dried dung (g) | | Ash free organic matter (% of dry weight) | |
|---|---|---|---|---|---|
| | | Exclusion cage | Uncaged | Exclusion cage | Uncaged |
| Early season | 2 | 168.09 ± 3.63 Aa | 116.57 ± 9.19 Ab | 87.12 ± 1.21 Aa | 79.20 ± 0.48 Ab |
| | 4 | 125.12 ± 6.06 Ba | 80.83 ± 5.27 Bb | 86.27 ± 1.49 ABa | 78.52 ± 1.12 Ab |
| | 7 | 114.67 ± 1.90 BCa | 92.32 ± 5.41 Bb | 84.65 ± 0.60 ABa | 78.01 ± 0.53 ABb |
| | 14 | 133.28 ± 3.11 Ba | 79.51 ± 8.08 BCb | 84.93 ± 0.69 ABa | 76.60 ± 0.65 Bb |
| | 28 | 114.51 ± 3.07 BCa | 74.13 ± 5.63 BCb | 84.33 ± 0.27 Ba | 74.61 ± 0.50 Ca |
| | 42 | 92.57 ± 1.72 Da | 58.67 ± 2.32 Db | 80.21 ± 0.53 Ca | 69.70 ± 0.23 Db |
| | | exclusion: $F_{1,60} = 182.80, P < 0.001$ time: $F_{5,60} = 34.55, P < 0.001$ interaction: $F_{5,60} = 2.45, P = 0.044$ | | exclusion: $F_{1,60} = 350.22, P < 0.001$ time: $F_{5,60} = 28.11, P < 0.001$ interaction: $F_{5,60} = 1.60, P = 0.175$ | |
| Late season | 2 | 267.33 ± 6.44 Aa | 204.23 ± 3.92 Ab | 88.58 ± 0.59 Aa | 85.91 ± 0.82 Ab |
| | 4 | 190.59 ± 5.55 Ba | 158.49 ± 1.50 Bb | 87.59 ± 0.53 ABa | 84.87 ± 0.80 ABb |
| | 7 | 192.65 ± 1.83 Ba | 158.61 ± 2.49 Bb | 87.17 ± 0.46 ABCa | 84.75 ± 0.96 ABa |
| | 14 | 197.65 ± 6.51 Ba | 141.99 ± 2.91 Cb | 86.95 ± 0.60 BCa | 83.61 ± 0.85 Bb |
| | 28 | 164.24 ± 2.55 Ca | 94.55 ± 3.17 Db | 85.84 ± 0.42 Ca | 81.76 ± 0.89 Cb |
| | 42 | 155.66 ± 2.85 Ca | 93.63 ± 3.61 Db | 85.88 ± 0.38 Ca | 79.88 ± 0.47 Db |
| | | exclusion: $F_{1,60} = 531.25, P < 0.001$ time: $F_{5,60} = 205.94, P < 0.001$ interaction: $F_{5,60} = 8.23, P < 0.001$ | | exclusion: $F_{1,60} = 144.60, P < 0.001$ time: $F_{5,60} = 20.26, P < 0.001$ interaction: $F_{5,60} = 3.49, P < 0.001$ | |

than the caged pats. Initially, 90.26% of dried dung pats were AFOM by weight, and by the 42nd day it decreased to 69–80% for uncaged pats and 80–86% for caged pats. Early season uncaged dung pats averaged 6.8% less AFOM than caged pats, and late season pats followed a similar trend with 3.14% less AFOM in the uncaged compared to the caged pats. Early and late season AFOM weight was also significantly affected by exclusion cages and time (Table 1), and the loss of AFOM weight was different between treatments (Fig. 3). Early season dung pats with arthropods are estimated to completely degrade before 71 d and this increases to 104 d when arthropods are excluded from the pats with cages. Late season dung pats achieve complete breakdown at a faster rate with estimates of 61 d and 100 d in uncaged and caged pats respectively.

### Effect of dung beetles on arthropod community

Although they represented 1.5–3% of the arthropod community recovered, dung beetle abundance, diversity, and richness were always positively correlated with arthropod community characteristics (Table 2). The abundance of dung beetles was significantly and positively correlated with total arthropod biomass, arthropod abundance, total species richness, and abundance of the micro-coprophage community in both the early and late seasons. Dung beetle species richness and diversity was correlated to an increase in abundance of the entire dung arthropod community in both early and late seasons (Table 2) (Fig. 4).

## DISCUSSION

Dung degraded more quickly when all arthropods were allowed access to the dung pat. Dung pat wet weight, dry weight, moisture percentage, and AFOM all decreased over time during the 42-d observation period (Table 1, Fig. 3). The degradation characteristics also show that uncaged pats degraded faster than the caged ones. Ash-free organic matter (AFOM) has been proposed as the most accurate measure of dung pat degradation (*Holter, 1979*). When arthropods were allowed to colonize dung pats, AFOM was reduced substantially within the first 2 d (Fig. 3). This quicker degradation may be explained by early colonization of the pat by relatively large arthropods. Dung beetles are some of the largest dung arthropods, and the amount of dung they consume and remove from the pat for oviposition is disproportionate compared to their abundance in the dung pats (*McDaniel, Boddicker & Balsbaugh, 1971*). After this first 2 d, the pats degraded at similar rates in both the caged and uncaged pats. When insects were allowed access to the pats, the pats had 32% of the original AFOM after 42 d; caged pats had 55% of the AFOM at the end of the observation period. Extrapolations of the data show that insects shorten the lives of pats (i.e., complete AFOM removal) by an estimated 33 d in the early season, and by 38 d later in the season. These observations are comparable to dung degradation estimates made in similar studies (*Lee & Wall, 2006*; *Tixier, Lumaret & Sullivan, 2015*). Often these examples only exclude arthropods for short periods of time or focus on the collection of a single arthropod group, limiting the scale and scope of the observations made. Results show that early dung pat degradation sets the tone for the remainder of the dung pat's time on the soil surface (i.e., the slopes of the regressions presented in Fig. 3). This suggests that

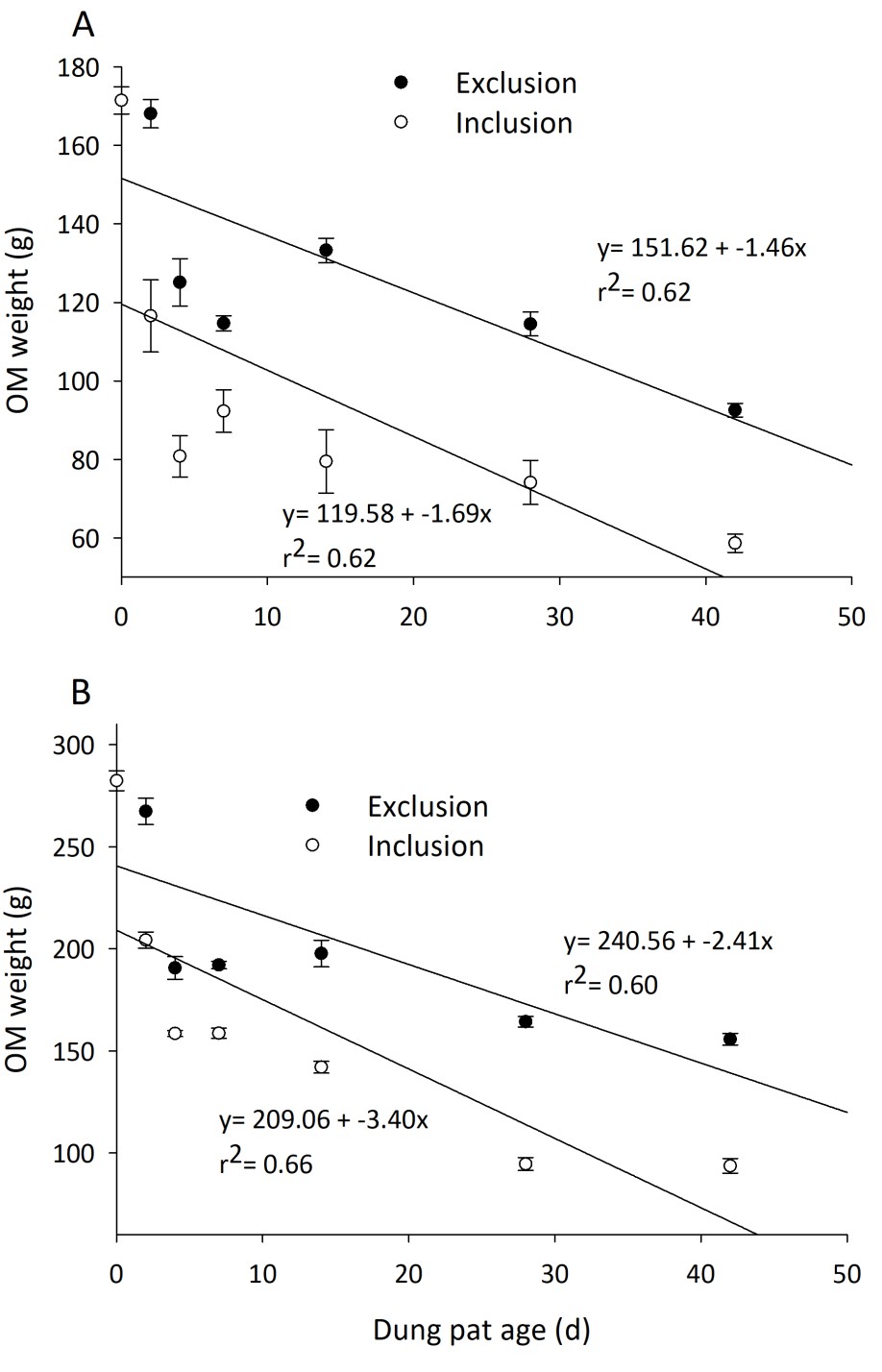

**Figure 3  Degradation rate of organic matter content (mean ± SEM) in cattle dung pats.** "Exclusion" refers to caged pats; "Inclusion" refers to uncaged pats. Dung pats were dried to 0% moisture and burned in furnace to remove all organic matter allowing calculation of ash-free organic matter content (AFOM). AFOM content was calculated in dung pats beginning in June (A) and late July (B). Half of pats had arthropods excluded ($n = 6$ pats per treatment per dung age) compared to allowing uninhibited arthropod colonization.

**Table 2  Relationships of dung beetle abundance, richness, and diversity to the arthropod community characteristics when complete insect communities were allowed access to dung pats.** Communities were sampled early in the season and late in the season, and are presented distinctly. Data presented represents the mean $\pm$ SEM. Statistical presentation are the result of linear regressions, and $\alpha = 0.05$.

| | Characteristic | Dung beetle | | |
| --- | --- | --- | --- | --- |
| | | Abundance | Richness | Diversity |
| Early season | Arthropod biomass (mg) | $F_{1,35} = 47.64, P < 0.001$ | $F_{1,35} = 74.46, P < 0.001$ | $F_{1,35} = 59.42, P < 0.001$ |
| | Arthropod abundance | $F_{1,35} = 44.84, P < 0.001$ | $F_{1,35} = 15.07, P < 0.001$ | $F_{1,35} = 10.73, P = 0.001$ |
| | Species richness | $F_{1,35} = 42.53, P < 0.001$ | $F_{1,35} = 74.46, P < 0.001$ | $F_{1,35} = 16.31, P < 0.001$ |
| | Micro-coprophage abundance | $F_{1,35} = 52.80, P < 0.001$ | $F_{1,35} = 19.69, P < 0.001$ | $F_{1,35} = 15.16, P < 0.001$ |
| Late season | Arthropod biomass (mg) | $F_{1,34} = 70.80, P < 0.001$ | $F_{1,34} = 63.94, P < 0.001$ | $F_{1,34} = 61.20, P < 0.001$ |
| | Arthropod abundance | $F_{1,34} = 27.23, P < 0.001$ | $F_{1,34} = 30.00, P < 0.001$ | $F_{1,34} = 27.54, P < 0.001$ |
| | Species richness | $F_{1,34} = 42.95, P < 0.001$ | $F_{1,34} = 63.94, P < 0.001$ | $F_{1,34} = 81.97, P < 0.001$ |
| | Micro-coprophage richness | $F_{1,34} = 25.06, P < 0.001$ | $F_{1,34} = 28.65, P < 0.001$ | $F_{1,34} = 28.31, P < 0.001$ |

even a short period of exclusion could have implications for the degradation of a dung pat. Disruptions to early arthropod colonization can have long-term implications to the efficient recycling of dung pats.

With few exceptions, the cages effectively reduced both the diversity and abundance of arthropods in cattle dung pats. The arthropods found in caged dung pats were those existing in the surrounding soil and those species that were small enough to fit through the exclusion screen (e.g., flies laid their eggs on the screen, and neonate larvae fell through onto the dung pat). There was significantly higher arthropod abundance and biomass found in the uncaged pats that were younger than 7 d old (Figs. 1 and 2). These significantly higher numbers of early colonizers corroborate previous work that showed that the highest densities of invertebrates occur between 2 and 5 d post-deposition (*Kessler & Balsbaugh, 1972*; *Lee & Wall, 2006*). Most of the arthropods found in the caged dung pats were small hydrophilid beetles, ptiliid beetles, mites, and Collembola, which collectively are described as "micro-coprophages". This group frequently colonized both caged and uncaged pats due to their small size and presence in the soil prior to cage placement. Another group found in both treatments was dipteran larvae; indeed, we found more maggots in the caged pats than the uncaged pats. Overall, there was a significant and substantial reduction in the biomass, abundance and diversity of most the dung arthropods in the cages.

Arthropod community complexity and abundance diminished as the pat aged past 7 days. Many of the early dung colonizers (flies and coprophagous beetles) consume small particles found in the liquid portion of freshly excreted pats (*Holter & Scholtz, 2007*). The offspring of these early dung pat colonizers add complexity to the community. Once sufficient numbers of these coprophages and their larvae have aggregated in the dung pat, a wave of predatory arthropods respond to this prey source (*Koskela & Hanski, 1977*). As the moisture evaporates from the dung, it becomes less attractive to many of the coprophages (*Stevenson & Dindal, 1987*). Additionally, many coprophages migrate to more recently deposited pats (*McDaniel, Boddicker & Balsbaugh, 1971*; *Mohr, 1943*). Predators follow these prey species (*Slade et al., 2016*; *Sowig, Himmelsbach & Himmelsbach, 1997*). This

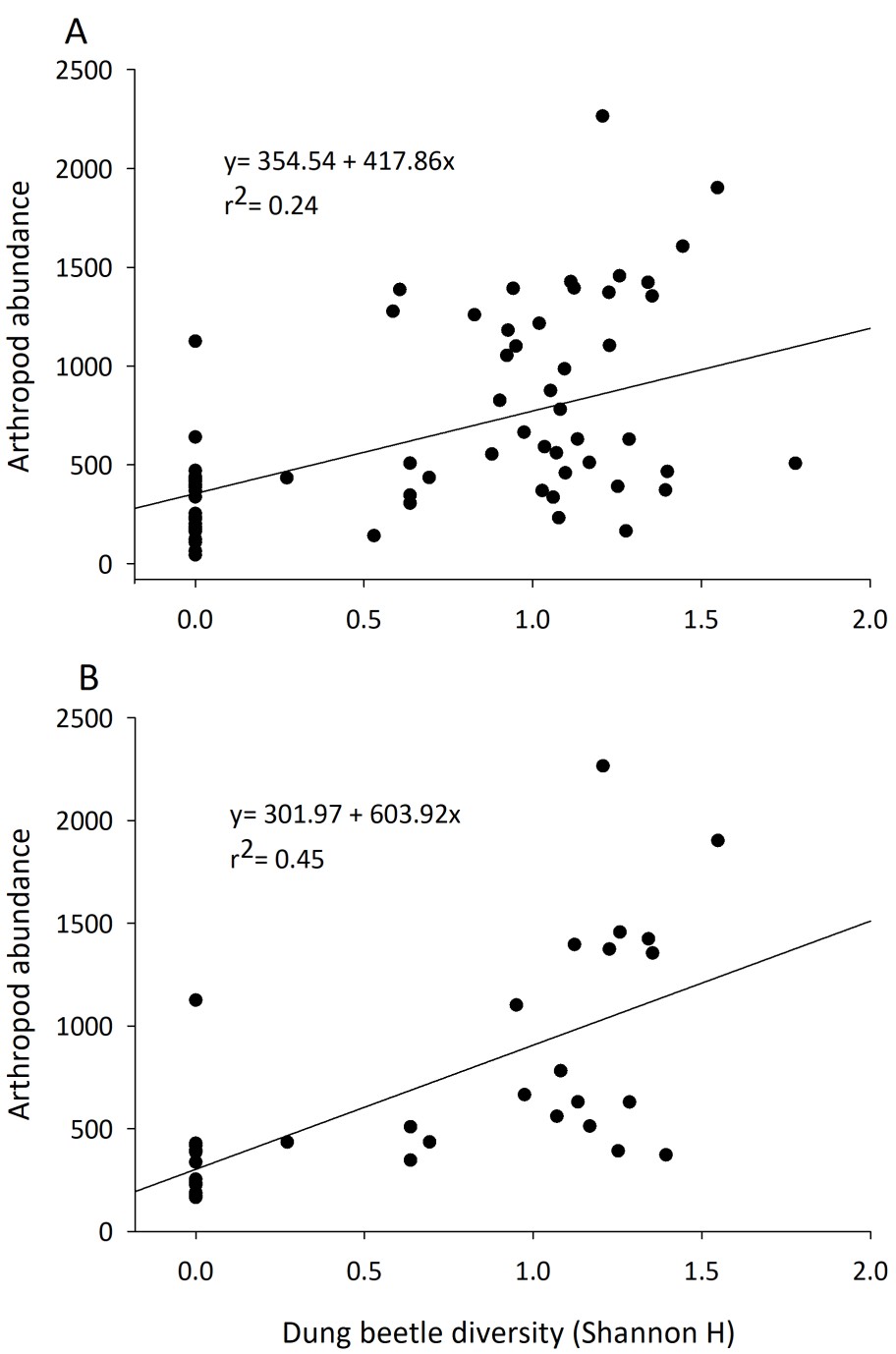

**Figure 4 Correlation of the diversity of dung beetles to total arthropod abundance in cattle dung pats.**
Dung beetle diversity (Shannon H) in dung pats was run in a linear regression to observe correlation to total arthropod abundance per individual cattle dung pat. Beginning in June (A) and late July (B) there was a significant and positive correlation between dung beetle diversity and the total arthropod abundance in dung pats that all species could freely colonize ($n = 36$ in both A and B).

succession of colonization is supported by observations of arthropods during this study. Arthropod abundance decreased as the pats lost moisture, and by the time dung was 14 d old, the arthropod community metrics and moisture content reached a constant low for the rest of the observation period. These observations resemble the succession of dung colonization seen in similar studies (*Kessler & Balsbaugh, 1972*; *Mohr, 1943*; *Valiela, 1969*).

In addition to their changes over the age of the pat, dung arthropod communities also change over the season. Due to the resource intensity of this study, it was conducted on a single ranch in a single year which challenges generalizations on seasonal patterns in dung degradations. Nevertheless, some trends between the two observation dates are noteworthy. In the early season, peak arthropod abundance was in 7 d old dung pats. Later in the season, peak abundance was in 2 to 4 d old pats, with a more gradual decrease in abundance as the pat aged (Fig. 2). Several explanations may factor into these patterns of dung colonization. Temperature has an important effect on dung colonization (*Errouissi et al., 2004*), with colder temperatures affecting colonization and degradation rates. In this study, early season had a colder temperature (16.7 °C daily average) than late season (21.1 °C daily average), and this may partially explain our experimental results. Additionally, many arthropods, such as dung beetles, do not share the same phenology, with many adults emerging and becoming active in different times over the grazing season (Pecenka and Lundgren 2018 in review). Higher temperatures later in the season would also dry the pat more quickly, and water content of the pat influences its attraction to dung arthropods (*Finn & Giller, 2000*). This higher temperature causes the dung pat to dry at a quicker rate, making it less attractive and suitable for dung beetles and other large coprophagous arthropods. As the grazing season progresses, cool season grasses are replaced with warm season species (*Ellis-Felege, Dixon & Wilson, 2013*). The changes in palatability and digestibility of different plants to cattle can affect the composition of the dung and its attractiveness to arthropods (*Holter, 2016*). Without further research, we cannot definitively say what drove these slightly different patterns of colonization over the season, but these considerations become important to ranchers wanting consistent dung degradation on their land.

These results provide further evidence that dung beetles contribute multiple ecosystem functions to rangelands by the dung arthropod community (*Beynon et al., 2012*; *Manning et al., 2016*; *Nichols et al., 2008*). Arthropod communities are major contributors to dung degradation, and dung beetle abundance and diversity influence many of the characteristics of this community (abundance, richness, and diversity) (Table 2, Fig. 4). Dung beetles were strongly correlated with the overall arthropod community even though they represented only 1.5–3% of the specimens collected in the study. Dung beetles colonize fresh dung pats and feed on the liquid portion of the dung; they leave when water becomes limited (*Holter & Scholtz, 2007*). Dung beetles may also deposit eggs in or underneath the dung pat, where their larvae will develop and consume the dried fibrous portion of the dung pat that remains (*Laurence, 1954*). Dung beetles can also alter the dung pat and the arthropods that will colonize it. Through their tunneling and bioturbation of the dung pat, they allow air to reach the center of the pat and cause it to degrade faster by converting it into forms accessible to plant roots and microbes (*Bang et al., 2005*; *Bornemissza, 1970*; *Stevenson & Dindal, 1987*). We hypothesize that dung beetles' robust bodies also provide a "highway

system'' that other arthropods such as predatory beetles or spiders can use to search for prey such as pest maggots. These tunnels would then also open up the pat's interior to the micro-coprophage community that lack the ability to burrow through the pat; further increasing their effect on pat degradation. Their impact can be seen in the large amount of OM lost in the first days of arthropod colonization.

## CONCLUSIONS

- Degradation of dung pats was increased by 30% (approximately 30 d) when the entire insect community was present in the pats.
- Early colonization was essential to dung degradation. Most dung degradation occurred within a week of dung deposition, and the main effects of insects on pat degradation occurred within 2 d of pat deposition.
- Although dung beetles were only 1.5–3.0% of the arthropod community in dung, their abundance was strongly correlated with the rest of the community. This data supports their role as essential contributors to dung degradation.

## ACKNOWLEDGEMENTS

The authors thank Mike Bredeson, Claire LaCanne, Amy Lieferman, Alexander Nikolas, Cassandra Pecenka, and Kassidy Weathers for their assistance in construction, deploying, and field removal of dung pat cages. Special thanks as well to Mark Longfellow in assisting with insect identification, and K Creek Ranch for use of their pasture.

### Funding

Ecdysis Foundation and NCR-SARE, via student research grant GNC15-207, funded this research. The funders had no role in study design, data collection and analysis, decision to publish, or preparation of the manuscript.

### Grant Disclosures

The following grant information was disclosed by the authors:
Ecdysis Foundation.
NCR-SARE: GNC15-207.

### Competing Interests

Jonathan Lundgren is the director of Ecdysis Foundation and CEO of Blue Dasher Farm.

### Author Contributions

- Jacob R. Pecenka conceived and designed the experiments, performed the experiments, analyzed the data, prepared figures and/or tables, authored or reviewed drafts of the paper, approved the final draft.
- Jonathan G. Lundgren conceived and designed the experiments, analyzed the data, contributed reagents/materials/analysis tools, prepared figures and/or tables, authored or reviewed drafts of the paper, approved the final draft.

## Data Availability

The raw data are provided in Data S1.

## Supplemental Information

Supplemental information for this article can be found online at http://dx.doi.org/10.7717/peerj.5220#supplemental-information.

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
