# Peer review of "The importance of dung beetles and arthropod communities on degradation of cattle dung pats in eastern South Dakota"

_PeerJ, doi:10.7717/peerj.5220_

## Round 0.1 · original submission · Minor Revisions

Both reviewers have provided extensive comments on your manuscript - both are positive towards your findings and have given you a great set of queries to really enhance the manuscripts value and impact. Please consider them all in your updated manuscript and clearly identify them in your rebuttal letter.

·

Basic reporting

I think this is quite an interesting study. The writing is professional, but in some cases is a little ambiguous and could use a little further attention-to-detail. The level of background information and context provided by the author is appropriate. The Tables use a particular notation to separate means that I find difficult to quickly interpret (use of two different cases, e.g. AaBd). In general, the results could be reported more succinctly and clearly. There are minor graphical errors (e.g. The error bar on the third groups of bars on Figure 2). There are some typos included in the Figure legends (e.g. Figure 3 – “Half of pats had arthropods excluded compared to allowing uninhibited arthropod colonization”).

It is unclear why means and error are shown instead of the raw data in Figure 3. The entirety of the raw data does not seem to be shared – e.g. it would not be possible to calculate species richness per sample based on the excel files provide by the authors. Reading the methods, I expect the measures of species richness and diversity may be a bit unreliable because different taxa are identified to varying levels of taxonomic resolution (141), and much of the richness of these samples will be cryptic (e.g. diptera larvae) and difficult to estimate. It would be helpful if the authors could justify this approach.

The data included in this publication appropriately constitute a single and comprehensive manuscript.

Experimental design

The research is original and falls within the aims and scope of PeerJ. The research question “This study uses an arthropod exclusion/inclusion system to evaluate the role of dung arthropods on dung degradation over time in eastern South Dakota… We hypothesize that an arthropod exclusion system will limit the colonization in the dung pats and result in a slower rate of pat degradation” would benefit from being described more succinctly.

The authors should ensure that their language reflects that their interest reflects the “consequence” of removing insects from cattle dung, and not the “process” or “means of”. The language that is currently being used is reflective of the latter.

The authors could make a stronger argument as to why e. South Dakota should be a focal/important region to ask these questions.

All evidence suggests that this is well-replicated study that is performed to a high technical and ethical standard. The authors would need to provide the entirety of the raw data (as mentioned above) to ensure their analysis is robust.

Validity of the findings

The authors have made multiple comparisons using the same data-set. It would be useful to include a brief discussion about the possibility of type 1 error when such an approach is used. There may be some issues encountered when assuming the data (e.g. insect abundance) is normally distributed, this should also be discussed.

The concluding thought of this paper was not as strong as the rest of the paper, and did not appropriately encapsulate the results of the study. To me, this is a crucial sentence to get right.

I am a little bit puzzled by the second part of this study (e.g. Table 2). This section does not seem to be as scientifically grounded as the first component. The experiment you describe does not seem to be set up to ask whether/how dung beetles 'drive' the overall insect/arthropod community. I suppose you could argue that dung beetles might serve as an indicator of these variables of interest because they are 'generally easy to identify and quantify'. However, I think the inclusion of this component distracts from your more interesting results (as reported in Table 1), and adds little value to the manuscript as a whole.

As I mention above, I think the authors would benefit by thinking carefully about how they present the results in Tables 1+2. Additional information within the Table legends might help this along. I find it quite overwhelming to look at, particularly the way you have shown the separation of the means. In addition, the authors should note what measure of error they are reporting.

Some of the discussion is oversimplified. For instance, Figure 2A shows that arthropod biomass in late summer was the highest at an intermediate time – which goes against the statement in the discussion “Arthropod community complexity and abundance diminished as the pat aged [L281]”.

Additional comments

Here are some smaller points that I thought might be useful to fix.
L82: See Beynon et. al 2015 for recent economic analysis.
96: Make it clearer that you believe that ‘excluding arthropods’ will be the factor that impairs/delays dung degradation, rather than the excluding system.
152: pseudo replication -> pseudoreplication
159: specimen ->specimens
197: this heading does not read as a complete thought. Please rephrase for clarity.
202: biomass(es) should not be pluralized here
218: please specify what measure of ‘average’
226: How was the community reduced? Diversity, abundance, richness? Please be more specific.
240-241: This result does not make sense, can you please clarify?
242: remind readers about what AFOM means. I forgot by the team I reached this point.
248: Sentences shouldn't usually begin with an acronym
261: was -> were
267: not totally clear you are talking about the arthropods in the ‘exclusion’ treatment
278: I think use of ‘caged’ vs ‘uncaged’ is much more intuitive than the terminology you use elsewhere in this paper.
306: watch ‘significant’ figures, hundredth of a C degree not important.
339: “for the duration of the pat’s life” -
332: Can you discuss the expedited decomposition provided by insects? The 33 days is relative to what? Would be more helpful as a measure of magnitude e.g. 33 days (10% reduction in time)
345: I am not convinced by your argument that ‘dung beetles were significant drives of the overall arthropod community? What does this mean? Can you give a little more context, and strengthen this argument?
357: To the best of my knowledge, I am not aware of any such studies that support this idea. Could you please include references?
359: This is not a terribly strong way of ending your paper. Try to end on a more conclusive note.

Lastly, you would benefit from a more detailed explanation of why you included the ‘sham cage’ treatment. You get into this a bit during the section beginning on L189, but there is room for improvement.

Reviewer 2 ·

Basic reporting

Do the authors plan to include the raw data of the species, species number, etc that they collected and identified? Or perhaps those data are in another manuscript?

Experimental design

The goals of the manuscript aren't particularly clear. Is the goal to examine dung degradation? In addition, the "excluded" treatment doesn't exclude arthropods, it just limits the body size of the arthropods that can enter. Thus, the study is really more about the impact of macrocoprophages in dung degradation as opposed to just microcoprophages -- which is very interesting. Why not focus the MS on this area instead of "excluded" even though arthropods weren't really excluded? There are few places where the authors could clarify methods (what is a sentinel pat, for example?).

Validity of the findings

As stated, the excluded treatments do not really exclude arthropods. Because of this, the authors cannot examine arthropod impacts in isolation despite this being the focus of the findings. See Experimental design section and comments to authors for more info on how this could be easily changed.

Additional comments

This study examines the role of arthropod communities in dung degradation on a South Dakotan cattle pasture field by excluding the majority of arthropods from experimental dung pats. The primary goals seem to be an examination of seasonal differences in degradation and arthropod community structure. This study provides an extensive survey of a wide range of arthropods present at dung pats and demonstrates that macro-coprophages play an important role in dung degradation. The introduction of the manuscript offers a particularly nice review of the literature on dung degradation and the role of arthropods in providing ecosystem services. The large number of adult and larval insects (87k!) that were identified to family and species is impressive. The manuscript is a pleasant read and provides interesting information. There are a few areas that could be improved upon or clarified prior to publication.

Big Picture Concerns
1. The methods used to collect arthropods likely underestimates the presence of dung beetles. As the authors spend a large portion of their discussion focusing on the role of these species, the authors should address the limitations of their methods in surveying dung beetle diversity. Specifically the authors should address the following:
a. By placing mesh screen underneath the dung pats, the authors may be limiting the behavior of tunneling dung beetles, which make up a large portion of pasture dung beetle communities. Dung beetle breeding behavior is crucial for their role in dung degradation because a good portion of the dung is moved underground. With the study designed used, adult dung beetles would be eating the dung, but they would not be tunneling. Therefore, degradation of dung is likely underestimated because dung that would have been moved underground is not being relocated.
b. Furthermore, dung beetles that roll dung away from the pat would not be located on the dung pat and would also be excluded from the authors’ collections. Therefore, the authors may be underestimating the abundance of rolling dung beetles.
2. The authors wrote “Overall, there was a significant and substantial reduction in the biomass, abundance and diversity of most the dung arthropods in the cages. This allows an isolation of the function of arthropods in degradation of cattle dung.” The last sentence really isn’t true. Micro-coprophages were in the exclosure treatments, so there was no “isolation of the function of arthropods” in this study. Indeed, macro-coprophages were absent, but not arthropods and not all coprophages. A better alternative might be to change the focus of the manuscript from treatments that “excluded” and “included” arthropods to a study of the impact of macro-coprophage exclusion on dung degradation. This is more in keeping with the data at hand and does not require any substantial reworking of the manuscript.
3. In the discussion, the authors mention that early colonization of pats by dung beetles may drive the differences in dung degradation between exclosure and open pats. The authors, though, do not present evidence that dung beetles are in higher abundances in the early days of dung deposition. This evidence is necessary to support authors’ claims about the importance of early colonizing dung beetles.
4. The overall goal of the study is not clear. Where the authors wrote “These ecosystem services provided by dung beetles (as well as other members of the dung arthropod community) have an economic value to the ranching operation, but most of the numbers used to generate these values are at least 37 years old (Beynon et al. 2012; Fincher 1981; Losey & Vaughan 2006).” Follow this with “Thus, our goal was to…” to specifically state the goal of the work.
5. In the introduction, the authors should consider discussing previous knowledge and research about variation in dung beetle communities across space and seasons. Previous research has worked to demonstrate why dung beetle communities vary temporally and spatially and what such variation means for dung degradation on both pasture and non-agricultural land. A discussion of these results is missing from the introduction, but is intricately linked with the authors’ statement that this research will demonstrate “how elements of this community affect degradation.”
6. In the introduction, the authors mention that the South Dakota pasture landscape has changed dramatically since previous arthropod studies were conducted. The authors should expand on this statement and explain how the landscape has changed. Is there more or less land being used for agriculture? Have the methods used for this agriculture changed? Is there less area remaining as tall grass prairie? These details are important for the ecology of the arthropod community, but are only minimally discussed by the authors.
7. In the methods, the authors should consider statistical analyses that describe the differences between the early season and late season experiments. The authors state that there were significant changes between the two seasons in their results section, but the statistics used for these analyses are missing from the methods portion.
8. Throughout the results and figures, the manuscript addresses both biomass and abundance of arthropods. The authors should clarify what these measures tell the reader about the arthropod community – what information do biomass and abundance independently provide? Otherwise, these should be combined as it seems repetitive to describe these results in different paragraphs and in different figures. If the authors choose to keep both figures, then consider clarifying why providing both biomass and abundance is important for the reader.
9. The big picture of this study is to examine the role of the arthropod community in degrading dung. The results addressing this big picture are those discussing the differences in dung degradation between the exclosure and open pats. The results discussing the differences in arthropod communities between exclosure and open pats simply tell the reader that the exclosure methods worked (to a degree, since not all arthropods were excluded). The big picture results, though, are buried by the less important result that the exclosures worked. Similarly, the discussion first addresses the differences in arthropod abundance in great detail before reaching the key discussion about differences in dung degradation. The authors should consider reorganizing their results and discussion sections to lead with dung degradation.
10. Regarding the analyses about the role of dung beetles in the arthropod community, the authors do not address the question of whether dung beetle abundance correlates with dung degradation, which seems to be the bigger picture question than the correlation between dung beetle abundance and overall arthropod abundance.
11. In figure 4, the authors do not describe why increases in dung beetle diversity should predict increases in abundance. Why explore this correlation?
12. The manuscript notes that “Two-way ANOVAs were used to investigate how dung pat age and exclusion cages affected dung pat and arthropod characteristics including pat wet weight, dry weight, moisture content, organic matter content, and arthropod biomass, abundance, species richness (number of morphospecies found), species diversity (Shannon H), and abundance of family Scarabaeidae.” In total, that is 18 different tests. Yet, the manuscript later includes the following: “The sham cages had similar arthropod communities and dung characteristics with the no cage treatment in 20 of the 22 statistical comparisons of different dung arthropod community groups and dung degradation metrics.” What are the 22 statistical tests?
13. The conclusions state “Degradation of dung pats was increased by 30% (approximately 30 d) by allowing insects access to pats.” However, in the caged experiments, insects did have access to pats. See comment 2 above.
14. Throughout the manuscript, the authors discuss the exploration of the arthropod community and dung degradation “across treatments” or “across sampling periods”. In these cases, “across” should be changed to “between” since there are only two sampling periods and two treatments.

Specific line items
L 52-53: The authors mention that “degradation” of the dung pat reduces nutrient availability, but it is unclear what “degradation” means in this sentence. Are the authors referring to the leaching of nutrients such as P and N? It would be clearer to define this term before using it in L 52-53 or reference it more clearly in L 50-51.
L 57: The authors should be more descriptive when describing that dung can take up to 3 years to degrade in other systems. Are the author’s describing systems other than agricultural land, systems experiencing other climate patterns, etc…?
L 78-79: It is unclear what types of species the authors are referring to as “predator and parasitoid species.” In the sentence’s context, it appears the authors are describing dung beetles, but dung beetles are not predators of pest larvae. The authors should clarify this statement and include a citation.
L 81-82: The paper does not attempt to generate an economic estimate of dung beetle ecosystem services. Since the authors are not filling this knowledge gap, this sentence feels out of place in the introduction. Furthermore, this sentence weakens the author’s argument about the importance of dung beetles on pasture land.
L 83: The authors should consider deleting “dung” before arthropods.
L114, 116: The description of “sentinel pats” is unclear. Is every pat placed in the field a sentinel pat or is this a special descriptor? Explain sentinel in more detail and describe how sentinel pats were used as opposed to non-sentinel pats.
L 141-143: This sentence is unclear. To clarify, the authors should replace “species were identified to at least the family level” to “specimens were identified to at least the family level.” If the authors are not identifying the specimens to the species level, it is unclear how the species are then identified to family level. There is also an error near the citation at the beginning of line 143 which confuses the meaning of this sentence – to clarify, change to “to species by Ratcliff…”
L 171: It is unclear what the numbers “14, 1, 12, and 13 morphospecies” refer to.
L 181-186: The authors do not describe the methods used to test whether the changes across the season were statistically significant. The authors should reserve the use of “significant” to statistical analyses.
L 226: delete “affected”
L 267: consider adding “the arthropods found in the exclosure treatments…”
L 295-296: The sentence beginning with “Although this study,” feels out of place. Throughout the rest of this paragraph, the authors are describing arthropod abundance. Ending this paragraph talking about dung degradation is not a clear, logical transition.
L370: The authors should revise their use of keystone species. The dung beetle community is diverse and made up of many species. Some of these species may be more important than others, so the whole community should not be labeled as a keystone species.
Table 1: Title of Table 1 isn’t exactly clear. Suggested changes are in parentheses: The effects of age (Days since dung deposition) on degradation of physical and chemical characteristics of dung pats with arthropod community exclusion/inclusion.
Table 2. Remove “throughout grazing season” from description. This is not throughout the season, but is rather two time points during the season. The title would therefore be: Relationships of dung beetle abundance, richness, and diversity to the arthropod community characteristics in dung pats from the inclusion treatment.

---

## Round 0.2 · accepted · Accept

The authors have done a good job in their revisions and rejoinder of reviewers comments. i am satisfied that the manuscript is acceptable for publication.

#